# 2-Ketogluconate Kinase from *Cupriavidus*
*necator* H16: Purification, Characterization, and Exploration of Its Substrate Specificity

**DOI:** 10.3390/molecules24132393

**Published:** 2019-06-28

**Authors:** Israel Sánchez-Moreno, Natalia Trachtmann, Sibel Ilhan, Virgil Hélaine, Marielle Lemaire, Christine Guérard-Hélaine, Georg A. Sprenger

**Affiliations:** 1Université Clermont Auvergne, CNRS, SIGMA Clermont, Institut de Chimie de Clermont-Ferrand, 63000 Clermont–Ferrand, France; 2University of Stuttgart, Institute of Microbiology, D-70569 Stuttgart, Germany

**Keywords:** 2-ketogluconate, 2-ketogluconate kinase, 2-ketogulonate, 2-keto-3-deoxygluconate, *Cupriavidus necator*, biocatalysis, monosaccharides phosphate

## Abstract

We have cloned, overexpressed, purified, and characterized a 2-ketogluconate kinase (2-dehydrogluconokinase, EC 2.7.1.13) from *Cupriavidus necator (Ralstonia eutropha)* H16. Exploration of its substrate specificity revealed that three ketoacids (2-keto-3-deoxy-d-gluconate, 2-keto-d-gulonate, and 2-keto-3-deoxy-d-gulonate) with structures close to the natural substrate (2-keto-d-gluconate) were successfully phosphorylated at an efficiency lower than or comparable to 2-ketogluconate, as depicted by the measured kinetic constant values. Eleven aldo and keto monosaccharides of different chain lengths and stereochemistries were also assayed but not found to be substrates. 2-ketogluconate-6-phosphate was synthesized at a preparative scale and was fully characterized for the first time.

## 1. Introduction

Rare ketoses have great potential, for instance, as chiral auxiliaries, as sweeteners, or (thanks to their biological properties) in pharmaceutical chemistry [1]. Among them, phosphorylated monosaccharides are of particular interest due to their central role in metabolic pathways [2,3]. Sugar phosphates, having a 2-keto functionality, can be produced by lyases or transferases. More precisely, they can be obtained by a variety of aldolases [4,5,6,7,8,9], a transaldolase [10], or a transketolase [5,11,12,13,14,15,16,17]. In vivo, phosphorylated monosaccharides are often obtained by direct phosphorylation of the corresponding monosaccharide, catalyzed by an ATP-dependent kinase. Such enzymes have also been efficiently applied for natural or unusual phosphorylated sugar preparation [2,3]. Kinases, as biocatalysts for the production of rare 2-ketoaldonate-phosphates, could also play a key role in the synthetic design of new biologically interesting compounds and enrich the arsenal of biocatalyst compounds. We turned to a bacterial 2-ketogluconate kinase (KGUK; EC 2.7.1.13.) as another and somewhat neglected biocatalyst for the formation of 2-ketoaldonate-6-phosphates. While chemical preparation of 2-ketogluconate-6-phosphate (KGP) has been described by vanadate/NaClO_3_-catalyzed synthesis [18], an enzymatic approach would be of interest, as nontoxic substances are used, and furthermore, it is done in sustainable conditions.

KGUK is involved in the glucose and 2-ketogluconate catabolism of several aerobic bacteria, but relatively few bacterial species are able to utilize 2-ketogluconate as the sole carbon source for growth and energy provision. Besides one Gram-positive *Leuconostoc mesenteroides* [19], the main 2-ketogluconate utilizers are Gram-negative proteobacteria such as *Pseudomonas*, *Aerobacter/Enterobacter/Klebsiella*, or *Ralstonia/Cupriavidus* species. These bacteria may grow directly on 2-ketogluconate, which is taken up by a specific transport system (KGUT) [20]. Intracellularly, the compound may then either be reduced to gluconate [21,22] or phosphorylated by an ATP-dependent 2-ketogluconate 6-kinase, KGUK [19,23,24,25,26,27] (Figure 1). Some aerobic bacteria such as *Pseudomonas putida*, *Pseudomonas aeruginosa*, or *Aerobacter* (now classified as either *Enterobacter* or *Klebsiella*) are able to convert glucose first to gluconate and subsequently to 2-ketogluconate by periplasmic PQQ-dependent dehydrogenases. 2-ketogluconate-6-phosphate is reduced to gluconate-6-phosphate and is then either degraded via the hexose monophosphate/pentose phosphate pathway or the Entner–Doudoroff (KDPG) pathway [20,26,28,29,30,31].

2-ketogluconate kinase was first discovered as an inducible activity in 1953 in *Aerobacter cloacae* [32] and then in *Pseudomonas fluorescens* [24]. The enzyme’s product, 2-ketogluconate-6-phosphate, was isolated and described at the same time [23]. Later, four other KGUKs were identified from: (i) the Gram-positive *L*. *mesenteroides* [19], (ii) *Aerobacter aerogenes* (nowadays classified as *Klebsiella* pneumoniae) [25], (iii) *Hydrogenomonas eutropha* H16 (newer and alternative designations are *Ralstonia eutropha* H16 or *Cupriavidus necator* H16) [33], and (iv) *P*. *aeruginosa* [20,28]. To the best of our knowledge, however, no studies on the substrate specificity of any bacterial KGUK have been published so far. We focused our attention on the kinase from *C*. *necator*, the complete genome sequence of which has been published [34], and we abbreviated this enzyme as KGUK*_Cnec_*. In this work, we cloned, overexpressed, and purified the recombinant N-terminal his-tagged 2-ketogluconate kinase from *C*. *necator* (KGUK*_Cnec_*) in *Escherichia coli*. For the first time, its substrate specificity was studied with different commercially available sugars and with various synthetic analogues of the natural substrate 2-ketogluconate. Finally, a preparative scale of the 2-ketogluconate-6-phosphate was performed to demonstrate the synthetic potential of this enzyme.

## 2. Results and Discussion

### 2.1. Cloning, Overexpression, Purification, and Characterization of KGUK from *C. necator*

The *kguK* gene from *C. necator* strain H16 was cloned by PCR amplification from chromosomal DNA. The protein matched the expected molecular weight of the cloned his-tagged KGUK*_Cnec_* (35.7 kDa). The analysis of the cell-free extract (CFE) showed good recombinant enzyme production (4800 U per liter) in the soluble fraction. Thanks to its attached 6-histidines tag, the enzyme could be easily purified by Immobilized Metal Affinity Chromatography (IMAC). Starting from 200 mL of expression cell culture (0.85 g of wet weight of cells after sedimentation), 20 mL of CFE were obtained (260 mg of protein with a specific activity of 0.38 U·mg^−1^). After IMAC purification, 4.7 mg of protein were obtained with a specific activity of 8.7 U·mg^−1^ (Table 1). Final yield of this purification method was 42% and the purification fold was increased to 22.8. The effect of imidazole from IMAC fractions on KGUK*_Cnec_* activity was evaluated. No activity differences were detected in samples before and after imidazole removal. Actually, imidazole displayed stabilizing properties in the KGUK*_Cnec_* activity during storage. Indeed, purified enzymes stored at 4 °C in the presence of 0.25 M of imidazole retained 90% of initial activity after one month, whereas protein samples in the absence of imidazole were totally inactive after only one night stored at 4 °C. The addition of possible stabilizers other than imidazole, such as BSA or glycerol, did not increase the stability. The effects of protein freezing and freeze-drying were also unsuccessful on enzyme stabilization. Consequently, IMAC fractions were directly stored after protein purification and imidazole was removed just before each experiment in order to avoid possible chemical interferences. No loss of activity was detected after the desalting procedure, so the specific activity of the final imidazole-free fraction remained the same after IMAC purification.

### 2.2. Enzyme Activity

The gene *kguK* from *C. necator* (NCBI Reference Sequence: YP_841324.1) encodes a putative 2-ketogluconate kinase enzyme (EC 2.7.1.13) in line with earlier biochemical work which had discovered such enzyme activity in the strain H16 (formerly termed *Hydrogenomonas*) [33]. 2-ketogluconate kinase activity was experimentally confirmed in the recombinant KGUK samples from IMAC purification, which showed a specific activity of 8.7 U/mg. Maximal enzyme activity was observed when a concentration of 1.25 mM of 2-keto-D-gluconate (KG) was employed in the activity assay. When the KG concentration was increased, a slow and continuous drop of the activity was observed (Figure 2A).

In addition, the effect of the ATP concentration on the enzyme activity was also examined. Maximum specific activity was found at ATP concentrations of 1.25 mM. However, when the concentration was increased over 5 mM, the enzyme activity drastically decreased, showing strong inhibition by substrate excess (Figure 3B). Kinase activity showed the Mg^2+^ requirement as an enzyme cofactor. As the real donor phosphate substrate is the complex Mg–ATP, it is crucial to use at least the same concentrations of Mg^2+^ as the ATP ones, in order to ensure the right enzyme activity in the assayed conditions. The maximal enzyme activity was found at Mg^2+^ concentrations of 5 mM when 1.25 mM of ATP was employed. This enzyme displayed a similar specific activity to that of the only one described from *A. aerogenes* (8.1 U/mg) [25].

### 2.3. Substrate Specificity

Substrate specificity of the KGUK from *C. necator* for the phosphate acceptors was studied on a broad variety of sugars with different chemical structures. Firstly, 11 commercially available aldo and keto sugars were tested: d-glucose, l-glucose, d-fructose, d-psicose, d-tagatose, d-ribulose, d-xylulose, d-sorbose, l-sorbose, d-erythrose, and 2-deoxy-d-ribose, where both chain length and stereochemistry were varied. These compounds were reacted with KGUK and reaction progress was followed by the described spectrometric assay to assess enzyme activity (See Materials and Methods section). None of the 11 selected sugars showed any conversion, revealing they are not substrates for KGUK*_Cnec_* in our experimental conditions.

The study was then focused on substrates with closer chemical structures to the natural KG (i.e., KGul and KGal). In addition, KGUK specificity for 3-deoxy analogues was also examined (Figure 4). Indeed, KGUK*_Cnec_* displayed some amino acid sequence identity with previously described 2-keto-3-deoxy-d-gluconate kinases (KDGK) belonging to a different kinase family (EC 2.7.1.45) (Figure 5).

Thus, KDGK from *Thermus thermophilus* displayed 35.9% amino acid sequence identity with the KGUK from *C. necator*. KDGK catalyzes the ATP-dependent phosphorylation of KDG (Figure 6), with KDG being the C3 deoxy analogue of KG. Enzymes from this family were also described to be able to catalyze the phosphorylation reaction of KG [35,36]. Nevertheless, there are no data in the literature about KDG as a substrate of KGUKs, so we decided to explore the KGUK*_Cnec_* activity also using KDG and a C4 epimer (KDGul) as substrates.

KGul, KGal, and KDGul were prepared as recently published [37] by using pyruvate aldolases discovered from biodiversity. They were found to be able to use hydroxypyruvate and d-glyceraldehyde as nucleophile and electrophile substrates, respectively. In order to evaluate the catalytic properties of the KGUK*_Cnec_* toward the five obtained compounds (Figure 4), the kinetic parameters of the enzyme were calculated (Figure 2B–D). Kinetic parameters for the donor substrate ATP were evaluated as well, and the results are summarized in Table 2.

The kinetic parameters showed that KDG is a substrate of KGUK but with less efficiency than KG (*k*_cat_/K_M_ = 7370 and 16,770 s^−1^M^−1^, respectively). This was the opposite of what was observed in the KDGK enzymes: Kinase activity using both KDG and KG as substrates has been described in KDGKs from *Sulfolobus tokodaii* and *T. thermophilus* and, in both cases, KDG was the best substrate, whereas KG phosphorylation was less efficient [35,36]. KGul and KDGul were found as new substrates for KGUK*_Cnec_*, although they were converted with lower catalytic efficiencies than KG and KDG. On the other hand, KGal gave no reaction. KG appeared clearly as the best substrate (*k*_cat_/K_M_ = 16,770 s^−1^·M^−1^), whereas its epimer on C3 and C4 (KGul) reacted 60-fold slower (*k*_cat_/K_M_ = 246 s^−1^·M^−1^), revealing the importance of the stereochemistry (3*S*, 4*R*) within the active site. When a hydroxy group in C3 was missing (KDG), the enzyme maintained good efficiency, as evidenced by the decrease of only a half order of magnitude (*k*_cat_/K_M_ = 7370 s^−1^·M^−1^). Nevertheless, without a hydroxy moiety in the third position, as well as using the epimer of KDG in C4 (KDGul), a drastic decrease in efficiency (*k*_cat_/K_M_ = 83 s^−1^·M^−1^) was observed. Thus, configuration in C4 seems to be very important for KGUK activity.

### 2.4. Synthesis of 2-ketogluconate-6-phosphate

KGUK*_Cnec_* was used as a biocatalyst to prepare KGP at a preparative scale as the key product for metabolic studies. Thus, a biocatalytic system based on phosphorylation of KG with this new enzyme was developed. The reaction was first assayed on a small scale in order to find out the optimal conditions. An ATP regeneration system based on phosphoenolpyruvate (PEP)/pyruvate kinase (PK) system was implemented (Figure 7B) to avoid both the difficulty in separating ADP from KGP and to circumvent inhibition by ATP at [ATP] > 5 mM (Figure 3B). Indeed, the PEP/PK regeneration system has been proved to be compatible with a one-step purification of phosphorylated sugars via their precipitation as Ba^2+^ salts [7]. Reaction progress was monitored by measuring pyruvate formation during ATP regeneration. The reaction was optimized by varying the concentrations of KG and PEP in Tris-HCl buffer (1.0 mL, 50 mM, pH 8.0), containing catalytic amounts of ATP (2.5 mM) and MgSO_4_ (4 mM), in the presence of KGUK*_Cnec_* (0.35 U) and PK (1.7 U). KG and PEP were used in a maximum concentration of 50 mM.

Four different KG/PEP substrate ratios were assayed for determining the optimal concentrations. PEP was used as either the limiting substrate (KG/PEP: 1.0/0.5, 1.0/0.7, and 1.0/0.9) or in excess (KG/PEP: 1.0/1.1). In all cases, final KGP accumulation was lower than 70%. The best yield (Figure 8A) was obtained with 1.0/0.5 as the KG/PEP ratio (70% of KGP accumulated after 3 h of reaction). Optimization reactions were continued by increasing the final volume (2 mL) and decreasing the concentration of the limiting substrate (20 mM). In these new conditions, the best results (Figure 8B) were found with a KG/PEP ratio of 1.0/0.7 (80% of KGP accumulation after 6 h). Due to the positive effect observed during dilution, a third reaction was finally implemented in a final volume of 2.5 mL with 14 mM of KG and a KG/PEP ratio of 1.0/0.8. The reaction was performed at room temperature and, under these latter conditions, the phosphorylated compound accumulation in the reaction media reached 100% (Figure 8C) after overnight gentle stirring (100–200 rpm) (approx. 12 h).

Thus, KGP could be obtained in 85% yield of pure barium salt product, corresponding to a 0.6 g scale. A single precipitation of KGP directly from the reaction mixture as its barium salt led to pure KGP, as depicted by the ^1^H as well as ^13^C NMR spectra available in the Materials and Methods section. Importantly, 2-ketogluconate-6-phosphate was for the first time fully characterized. Indeed, although this biocatalytic approach had previously been used for KGP synthesis [23,25], this product was only identified by TLC then.

## 3. Materials and Methods

### 3.1. General Remarks

Protein analysis by SDS-PAGE was performed using 15% and 5% acrylamide in the resolving and stacking gels, respectively. Gels were stained with Coomassie brilliant blue R-250 (Sigma-Aldrich). Electrophoresis was run under reducing conditions in the presence of 5% β-mercaptoethanol. Pyruvate kinase (PK), lactate dehydrogenase (LDH), ATP, and 2-keto-D-gluconate (KG) were purchased from Sigma-Aldrich. 2-keto-3-deoxygluconate (KDG) was prepared as described by Lamble et al. [38] using a pyruvate aldolase from *Sulfolobus solfataricus*. 2-ketogulonate (KGul), 2-keto-3-deoxygulonate (KDGul), and 2-keto-d-galactonate (KGal) were prepared as described by de Berardinis et al. [37]. *E. coli* BL21(DE3) pLysS competent cells and pET28a expression vector were purchased from Invitrogen. Plasmid DNA purification kits were from Sigma-Aldrich. All other chemicals were purchased from Sigma-Aldrich as reagent grade.

### 3.2. Methods

^1^H (400 MHz) and ^13^C (100 MHz) nuclear magnetic resonance (NMR) analyses were carried out with a Bruker Avance 400 MHz spectrometer. Mass spectra were recorded on a Q-exactive spectrometer from Thermos Scientific using an electrospray ionization (ESI).

#### 3.2.1. Cloning

To amplify the *kguK* gene, chromosomal DNA from *C. necator* (*R. eutropha*) H16 (strain donated by Dr. Dieter Jendrossek, IMB, Univ. Stuttgart) was used. Primers for PCR with PwoI DNA polymerase were kguK-NdeI (5´ TTTTCATATGAGCACCGATCTTGACGTGG 3´, engineered NdeI site underlined) and kguK-BamHI (5´ TTTTGGATCCTCACAAACTGGCGGCCGC 3´, engineered BamHI site underlined). The amplified DNA was cut with NdeI and BamHI and ligated to a likewise cut pBluescriptSK vector (Agilent Technologies). The ligation mixture was then used to transform *E. coli* DH5α cells on LB-ampicillin plates (Amp 100 mg/L) with X-Gal (blue-white selection). White colonies were analyzed for correct insertion. The *kguK*-containing NdeI-BamHI fragment was then cloned into a likewise cut pET28a(+) vector (Invitrogen) with selection for kanamycin resistance. The presence of the cloned gene was verified by custom DNA sequencing (GATC Biotech, Konstanz, Germany). The vector pET28a-kguK*_Cnec_* thus encoded a N-terminal His_6_-tag fused to the protein (for expression as N-terminally his-tagged protein in order to simplify its purification procedure by IMAC).

#### 3.2.2. Expression and Purification

The gene expression was done in *E. coli* BL21(DE3) pLysS with induction by IPTG. Expression of KGUK*_Cnec_* in BL21(DE3) pLysS cells was evaluated by SDS-PAGE. Colonies containing the plasmids pET28a-kguK*_Cnec_* were cultured in Luria–Bertani (LB) broth in the presence of kanamycin (30 mg/L) as a selection antibiotic at 37 °C under orbital shaking (200 rpm). When the culture reached an OD_600nm_ of 0.5, protein expression was induced by adding IPTG (0.5 mM final concentration), and the temperature was lowered to 30 °C. The culture was incubated for a further period of 12 h. Cells were harvested by centrifugation, washed twice, and resuspended in buffer A (50 mM NaH_2_PO_4_, 300 mM NaCl, pH 8.0). Cell suspension was disrupted by ultrasonication and the cell lysate was centrifuged at 10,000× *g* for 20 min. Clear supernatant (20 mL) was loaded onto a Ni^2+^-NTA-agarose resin column (Qiagen, h = 1.5 cm; Ø = 2.5 cm) pre-equilibrated with buffer B (buffer A plus imidazole 20 mM). The column was washed with buffer B, and the retained proteins were eluted with the same buffer containing imidazole at a concentration of 250 mM. Eluted fractions containing pure protein were pooled together and directly stored at 4 °C. In order to avoid possible interferences, imidazole was removed before each enzymatic experiment using a desalting column system (PD-10 sephadex G-25M columns, Pharmacia) pre-equilibrated with a buffer of 50 mM NaH_2_PO_4_, pH 8.0 (final buffer). The imidazole removal was carried out just before each experiment by loading 2 mL of IMAC purified enzyme into the G-25M columns, equilibrated with the final buffer. The elution fractions containing the enzyme (2 or 3 mL) were pooled together and its specific activity was assayed. No loss of activity was detected after the desalting procedure.

#### 3.2.3. Enzyme Activity Assays and Kinetic Studies

Phosphorylation reactions involving different substrates catalyzed by KGUK were spectrophotometrically evaluated by measuring the release of ADP using a coupled assay with PK and LDH (Figure 7A) [39].

A typical assay was performed in a 1 mL reaction mixture containing Tris-HCl (25 mM, pH 8.0), NADH (0.2 mM), ATP (1.25 mM), phosphoenolpyruvate (PEP, 1.0 mM), MgSO_4_ (5.0 mM), KCl (50 mM), KGUK*_Cnec_* (2.0–10.0 μg), PK (3.3 U), LDH (2.3 U), and 1.25 mM of KG. Similar activity assays were used to evaluate the KGUK*_Cnec_* specificity for different phosphate acceptors, where KG was replaced by the corresponding analyzed substrate. One unit of kinase activity was defined as the amount of enzyme able to produce 1 μmole of 2-ketogluconate-6-phosphate (KGP) per min under the above conditions, using KG as the substrate.

Assays to determine kinetic parameters were performed following the kinase activity at different substrate concentrations under the general conditions described above. Steady-state kinetic assays for kinase activity were measured at 25 °C in a total volume of 1 mL. Measurement of kinetic parameters for ATP was carried out with 2.1 μg/mL of purified KGUK*_Cnec_* and a constant excess of Mg^2+^ of 5 mM in each assay point. KG was employed as the substrate (1.25 mM), and, as the phosphate donor, 14 different concentrations of ATP were used (Figure 3A). To evaluate the effect of higher concentrations of ATP on the enzyme activity, additional assays were carried out at a Mg^2+^ concentration of 25 mM to ensure the correct formation of the Mg–ATP complex (Figure 3B). In order to avoid ATP excess inhibition, the maximum ATP concentration used in assays for different phosphate acceptors was 1.25 mM in each kinetic point. Measurements of kinetic parameters for KG were performed with 2.5 μg/mL of purified KGUK*_Cnec_* at 15 different KG concentrations (Figure 2A). Assays to determine the kinetic parameters for KDG [38] were performed with 1.5 μg/mL of purified KGUK*_Cnec_* at 10 concentrations of the substrate (Figure 2B). Assays to determine the kinetic parameters for KGul and KDGul [37] were performed with 3.4 μg/mL of purified KGUK*_Cnec_* at 9 and 11 concentrations of the respective substrate (Figure 2C and D). Kinetic constants were calculated using the built-in nonlinear regression tools in the software SigmaPlot 12.0 (Systat Software Inc).

#### 3.2.4. Phosphorylation of 2-keto-d-gluconate.

##### Reaction Progress Monitoring

Phosphorylation reactions were followed by measuring the accumulation of pyruvate formed during the ATP regeneration process (pyruvate kinase/lactate dehydrogenase) using the spectrophotometric assay described above (enzyme activity, Figure 7A). Assays were performed in reaction mixtures of 1 mL containing the reaction aliquot Tris-HCl (40 mM, pH 8.0), NADH (0.2 μmole), and LDH (2 U). One millimole of oxidized NADH was equivalent to 1 mmole of pyruvate, which was equivalent to 1 mmole of KGP formed.

##### Preparative Scale Synthesis and Purification of 2-ketogluconate-6-phosphate

The optimal KG/PEP ratio was established to be 1.0/0.8 (Figure 8), where the phosphate donor was the limiting substrate. Using these optimal conditions, KGP was synthesized at a 1.5 mmole scale. Reaction mixture composition was set as follows: Tris-HCl buffer (104 mL, 50 mM, pH 8.0) containing PEP (1.5 mmole, 300 mg), KG (1.9 mmole, 400 mg), MgSO_4_ (4 mM), KCl (40 mM), KGUK*_Cnec_* (26.5 U), and PK (640 U). The reaction was initiated by addition of ATP (104 μmole, which was 20 times less than the limiting substrate). After 12 h of incubation, 100% of PEP (the limiting substrate) was consumed, which indicated a KGP accumulation of 1.5 mmole in the reaction medium.

The reaction mixture was quenched by dropping the pH to 3 by adding HCl (5 M), resulting in partial precipitation of the enzymes. The pH was then adjusted to 6 by adding NaOH (5 M), and 2 eq of BaCl_2_ dihydrate were added. The solution was centrifuged at 10,000 rpm at 4 °C for 10 min and the pellets were discarded. After partial concentration in vacuo, 5 volumes of ethanol were added. The solution was incubated overnight at 4 °C and then centrifuged. After one washing with ethanol followed by two other washings with acetone, KGP barium salt (molecular weight 476.5 g/mol) was recovered as a white powder in 85% yield (1.275 mmole, 0.608 g).

#### 3.2.5. Analysis

The sample existed under two cyclic forms: α and β pyranoses, the latter being the major one. Due to an overlap of the signals, it was difficult to precisely quantify each form.

^1^H NMR (400 MHz, deuterium oxide) δ 4.28 (d, *J* = 8.3 Hz, 1H, H3), 4.14 (t, *J* = 8.3 Hz, 1H, H4), 4.05–3.83 (m, 3H, H5 + H6).

^13^C NMR (101 MHz, deuterium oxide) δ 172.08 (βp, C1), 171.27 (αp, C1), 103.38 (αp, C2), 98.51 (βp, C2), 82.21 (αp, C3), 81.57 (d, *J* = 8.0 Hz, αp, C5), 79.55 (d, *J* = 8.4 Hz, βp, C5), 77.89 (βp, C3), 75.42 (αp, C4), 73.58 (βp, C4) 65.64 (d, *J* = 5.1 Hz, βp, C6), 65.23 (d, *J* = 4.6 Hz, αp, C6).

HRMS ESI-, m/z calcd. for [C_6_H_10_O_10_P] = 273.0012; found 273.0022.

^1^H NMR spectrum:



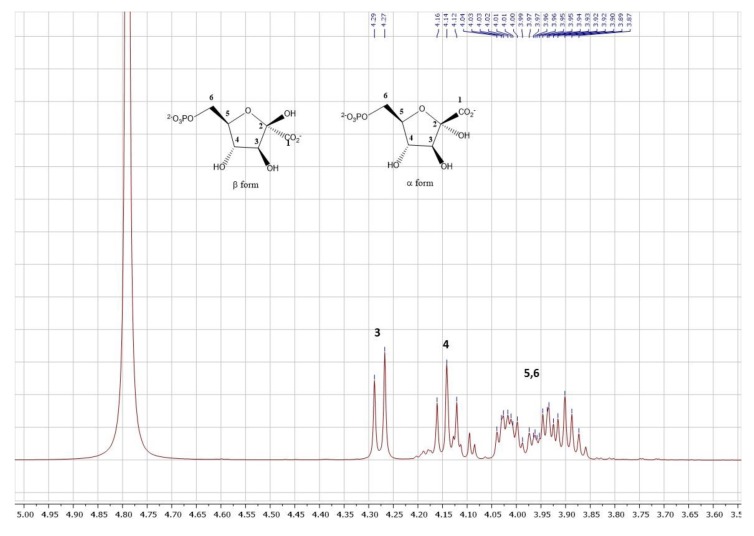



^13^C NMR spectrum:



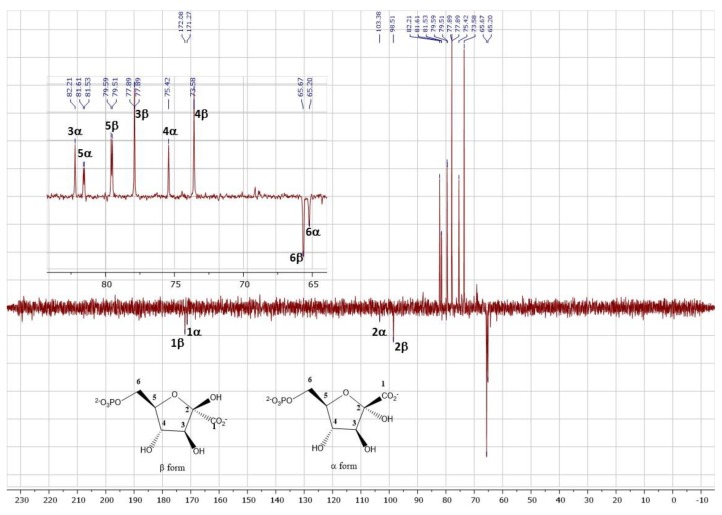



## 4. Conclusions

We successfully cloned, overexpressed, purified, and characterized KGUK from *C*. *necator* H16, which was first reported in 1974 but was never studied thereafter. This enzyme was found to be unstable in its pure form. We succeeded in stabilizing it by storage in an imidazole solution which was removed just before use. For the first time, we demonstrated that KDG is a substrate for this enzyme and that some KG and KDG epimers can also be converted into the corresponding phosphorylated derivatives. The ketoacid moiety was necessary since the complete set of keto or aldoses assayed were not found as substrates. Finally, KGP was successfully prepared at a preparative scale, with a good yield, and was fully characterized.

## Figures and Tables

**Figure 1 molecules-24-02393-f001:**
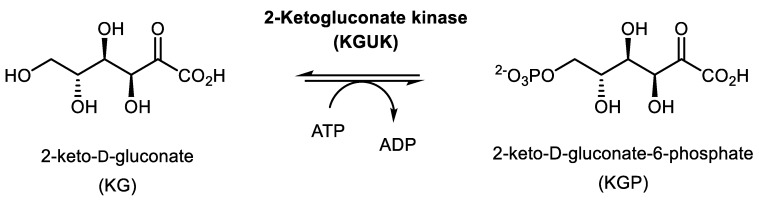
Reaction catalyzed by 2-ketogluconate kinase (KGUK).

**Figure 2 molecules-24-02393-f002:**
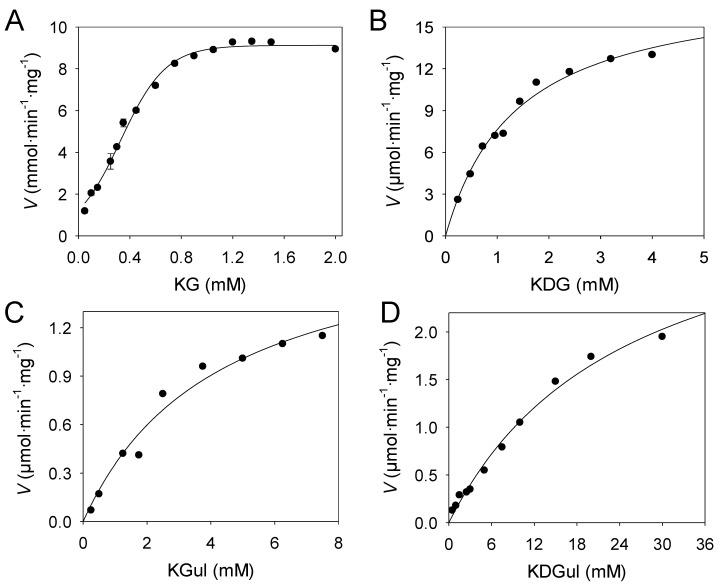
Substrate kinetics of kinase activity for KGUK from *C. necator* for different substrates (at 25 °C). Activity of purified KGUK was measured at increasing concentrations of the acceptor substrates 2-keto-D-gluconate (KG) (**A**), 2-keto-3-deoxygluconate (KDG) (**B**), 2-ketogulonate (KGul) (**C**), and 2-keto-3-deoxygulonate (KDGul) (**D**), maintaining a constant excess of ATP (1.25 mM) and Mg^2+^ (5 mM). Final concentration of pure KGUK was customized in each kinetic experiment to optimize the activity measurement: 2.5 μg/mL of purified enzyme for KG kinetics (**A**), 1.5 μg/mL for KDG assays (**B**), and 3.4 μg/mL for KGul and KDGul experiments (**C** and **D**).

**Figure 3 molecules-24-02393-f003:**
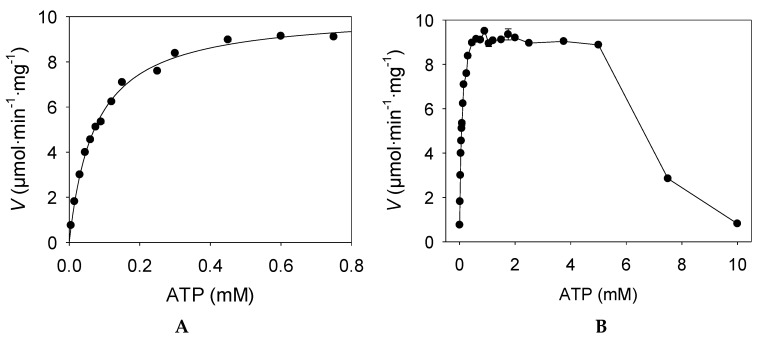
Kinase activity of the purified KGUK from *C. necator* was measured at increasing concentrations of ATP, maintaining a constant excess of KG (1.25 mM) as the phosphate acceptor and 2.1 μg/mL of purified enzyme. ATP kinetics were carried out for kinetic constants calculation using a Mg^2+^ excess concentration of 5 mM (**A**). To evaluate the effect of higher concentrations of ATP on the enzyme activity, additional assays were performed increasing the Mg^2+^ concentration to 25 mM (**B**). When ATP concentrations over 5 mM were used, a strong decrease in the kinase activity was observed.

**Figure 4 molecules-24-02393-f004:**
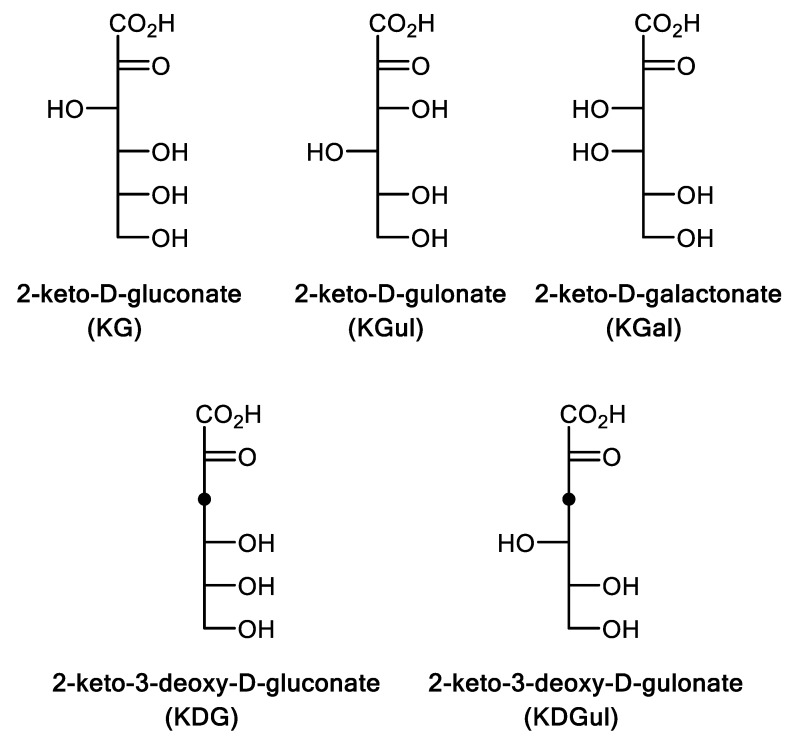
2-keto**-**carboxylic acid sugars tested as KGUK*_Cnec_* substrate.

**Figure 5 molecules-24-02393-f005:**
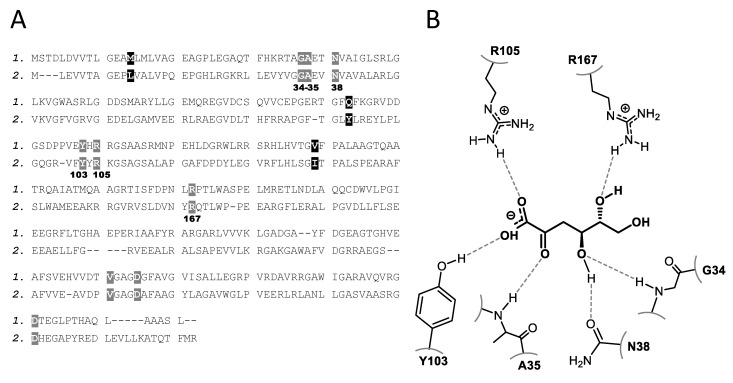
Sequence alignments and substrate accommodations of the concerned residues. (**A**) Amino acid sequence alignments of KGUK from *C. necator (1.)* and 2-keto-3-deoxy-d-gluconate kinase (KDGK) from *Thermus thermophilus (2.)*. Grey-shaded amino acids have been identified as active site residues in the crystallographic structure of the KDGK enzyme. Only three residues (shaded in black) are different in the *C. necator* enzyme. (**B**) Structure of the active site of KDGK from *T. thermophilus.*

**Figure 6 molecules-24-02393-f006:**
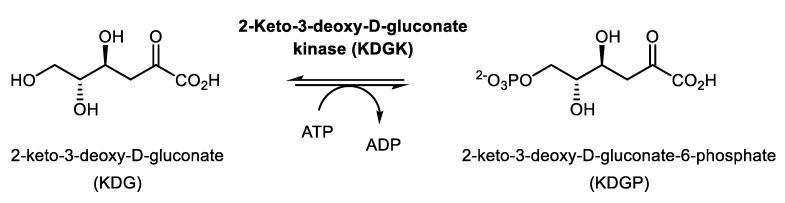
Reaction catalyzed by the KDGK enzymes.

**Figure 7 molecules-24-02393-f007:**
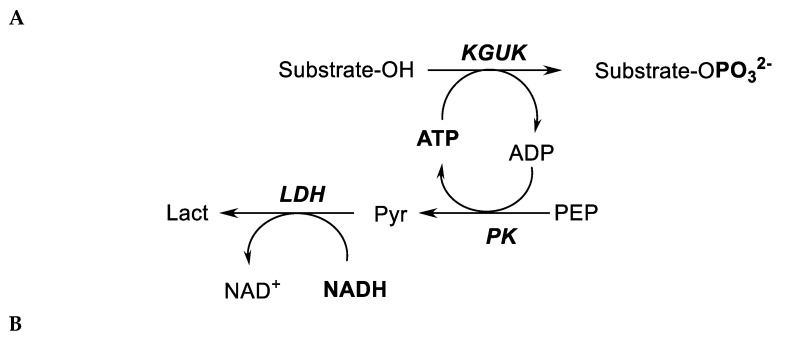
Phosphorylation reactions: analytical (assay) and practical (synthesis) scale. (**A**) KGUK activity assay. Substrate phosphorylation was measured with a coupled enzymatic system, where the decrease of NADH absorbance at 340 nm was directly proportional to substrate phosphorylation. PK: pyruvate kinase, PEP: phosphoenolpyruvate, pyr: pyruvate, lact: lactate, LDH: lactate dehydrogenase. (**B**) Biocatalytic orthogonal cascade for the synthesis of 2-ketogluconate-6-phosphate (KGP).

**Figure 8 molecules-24-02393-f008:**
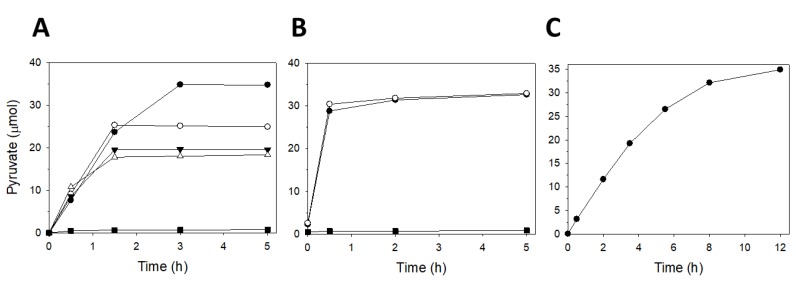
Optimization of the KG/PEP ratio in different KG phosphorylation reactions with ATP regeneration. (**A**) Reactions were performed in 1 mL of Tris-HCl buffer 50 mM and 50 μmol of the limiting substrate at different ratios of KG/PEP (1.0/0.5 (●), 1.0/0.7 (○), and 1.0/0.9 (▼), 1.0/1.1 (∆)) and a control reaction with no ATP (■). (**B**) Reactions were performed in 2 mL of Tris-HCl buffer 50 mM and 40 μmol of the limiting substrate at different ratios of KG/PEP (1.0/0.5 (●) and 1.0/0.7 (○)) and a control reaction with no ATP (■). (**C**) Reactions were performed in 2.5 mL of Tris-HCl buffer 50 mM and 35 μmol of the limiting substrate employing a KG/PEP ratio of 1.0/0.8.

**Table 1 molecules-24-02393-t001:** Summary of recombinant KGUK*_cnec_* purification (from 20 mL of cell-free extract (CFE)).

	Sample	Activity (U)	Protein (mg/mL)	Volume (mL)	Activity (U/mg)	Fold Purification	Recovery (%)
**1**	**CFE**	96.0	13.02	20	0.38	—	100
**2**	**IMAC**	40.4	0.47	10	8.70	22.8	42

**Table 2 molecules-24-02393-t002:** Kinetic constants of the recombinant KGUK*_cnec_*.

Entry	Substrate	V_max_ (U/mg)	*k*_cat_ (s^−1^)	K_M_ (mM)	*k*_cat_/K_M_ (s^−1^M^−1^)
**1**	**ATP**	10.2 ± 0.1	5.71 ± 0.05	0.073 ± 0.002	78,220 ± 2500
**2**	**KG^1^**	10.4 ± 0.4	5.8 ± 0.2	0.35 ± 0.03	16,770 ± 1300
**3**	**KGul**	1.9 ± 0.2	1.1 ± 0.1	4 ± 1	246 ± 70
**4**	**KDG**	18.2 ± 0.9	10.2 ± 0.5	1.4 ± 0.2	7370 ± 970
**5**	**KDGul**	3.7 ± 0.4	2.1 ± 0.2	25 ± 4	83 ± 20
**6**	**KGal***	-	-	-	-

^1^ KG substrate showed a sigmoid kinetics (see Figure 2A). Kinetic parameters were calculated by nonlinear regression in the Hill equation. Hill coefficient (n) = 1.4; K_M_ = K_0.5_. * KGal was not found to be a KGUK substrate under our assay condition.

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
