# Peer review of "2-Ketogluconate Kinase from Cupriavidus necator H16: Purification, Characterization, and Exploration of Its Substrate Specificity"

_molecules, 2019, doi:10.3390/molecules24132393_

Reviewer 1 Report

 Dear Authors,

 This should be deleted from results and discussion and reported only in material and methods

"(see Materials and Methods section) and subcloned into a pET28a(+) vector for expression as N-terminally his-tagged protein in order to simplify its purification procedure by IMAC. The gene expression was done in E. coli BL21(DE3) pLysS with induction by IPTG as described in Materials and Methods. Expression of KGUKCnec in BL21(DE3) pLysS cells was evaluated by SDS-PAGE."

Please rephrase this sentence “Its final production of 4,800 U per liter of CFE (cell-free extract) allowed protein purification by IMAC.” and remove “ allowed protein purification by IMAC.” As this is possible thanks to the His-tag.

 Please try to explain the reason of this behaviour (if possible of course!!) ” presence of 0.25 M of imidazole retained 90 % of initial activity after one month. Whereas protein samples in absence of imidazole were totally inactive after only one night stored at 4 °C.” (residual Ni in the eluted protein? )

did the Authors try adding EDTA instead of imidazole?

 This sentence is unfortunately not understandable ” KGul, KGal and KDGul were prepared as recently published [37] by using amazing pyruvate aldolases discovered from biodiversity which are able to use hydroxypyruvate and  D-glyceraldehyde as nucleophile and electrophile substrates, respectively.” Amazing?? Please rephrase into something meaningful.

Rephrase this as well “Kinetic parameters for the donor substrate ATP were evaluated as well, the results being summarised in and in Table 2. (“in and in” ? “results ARE summarised”)

 The conclusions must be improved. This sentence sounds a bit weird :” ….. but apparently was never studied thereafter.” Apparently??

 Best regards

Author Response

 Reviewer 1:

1-This should be deleted from results and discussion and reported only in material and methods

"(see Materials and Methods section) and subcloned into a pET28a(+) vector for expression as N-terminally his-tagged protein in order to simplify its purification procedure by IMAC. The gene expression was done in E. coli BL21(DE3) pLysS with induction by IPTG as described in Materials and Methods. Expression of KGUKCnec in BL21(DE3) pLysS cells was evaluated by SDS-PAGE."

This was done in the revised manuscript.

 2-Please rephrase this sentence “Its final production of 4,800 U per liter of CFE (cell-free extract) allowed protein purification by IMAC.” and remove “ allowed protein purification by IMAC.” As this is possible thanks to the His-tag.

Thank you very much for your comment. The sentence was rephrased as follows in the revised manuscript:

The analysis of the cell free extract (CFE) showed a good recombinant enzyme production (4,800 U per litre) in the soluble fraction. Thanks to its attached 6-histidines tag, the enzyme could be easily purified by IMAC.

 3-Please try to explain the reason of this behaviour (if possible of course!!) ” presence of 0.25 M of imidazole retained 90 % of initial activity after one month. Whereas protein samples in absence of imidazole were totally inactive after only one night stored at 4 °C.” (residual Ni in the eluted protein? ) did the Authors try adding EDTA instead of imidazole?

The most possible cause of this imidazole stabilizing effect is, as the Reviewer has pointed out, the presence of Ni2+ ions in the elution fractions: sometimes Ni2+ ions leak from the IMAC column and get stuck to the His-tag of the recombinant enzyme. Imidazole molecules in the elution media would chelate these Ni2+ ions thus removing the imidazole alone could make the protein insoluble. The best way to avoid this effect is to perform a dialysis in presence of EDTA but, unfortunately, we could not use this approach due to the strict dependence of the KGUK activity for Mg2+. Therefore, we decided to follow the strategy described in the manuscript: elution fractions from IMAC purification were perfectly stable due to the presence of imidazole. To avoid interferences in our enzyme assays, imidazole was removed by the desalting procedure just before the experiments were carried out.

We have highlighted this in the revised version stating on lines 101/102:: " No loss of activity was detected after the desalting procedure so specific activity of the final imidazole-free fraction remained the same than after IMAC purification."

and in chapter 3.2.2, last sentence by adding:

" The imidazole removal was carried out just before each experiment by loading 2 mL of IMAC purified enzyme into the G-25M columns, equilibrated with the final buffer. The elution fractions containing the enzyme (2 or 3 mL) were pooled together and its specific activity was assayed. No loss of activity was detected after the desalting procedure".

We hope that this clarifies the issue and thank the referee for its input.

4- This sentence is unfortunately not understandable ” KGul, KGal and KDGul were prepared as recently published [37] by using amazing pyruvate aldolases discovered from biodiversity which are able to use hydroxypyruvate and  D-glyceraldehyde as nucleophile and electrophile substrates, respectively.” Amazing?? Please rephrase into something meaningful.

Two sentences were rebuilt and the word amazing removed . The new sentence has been included: “KGul, KGal and KDGul were prepared as recently published [37] by using pyruvate aldolases discovered from biodiversity. They were found able to use hydroxypyruvate and d-glyceraldehyde as nucleophile and electrophile substrates, respectively.”

5- Rephrase this as well “Kinetic parameters for the donor substrate ATP were evaluated as well, the results being summarised in and in Table 2. (“in and in” ? “results ARE summarised”)

This was done.

6- The conclusions must be improved. This sentence sounds a bit weird :” ….. but apparently was never studied thereafter.” Apparently??

The conclusion was rephrased as follow:

“in 1974 and, however, was never studied thereafter.”

Reviewer 2 Report

Authors described expression and analysis of properties for KGUKCnec

There are many ambiguous points to understand in the whole of manuscript. Please recheck these ambiguous points and make revised manscript.

The major points are listed below.

 1)   Lane 102 ~ “after protein purification and imidazole was removed just before each experiment in order to avoid possible chemical interferences.”

Please describe typical specific activity (U/mg) after desalt procedure.

Many readers recalculated amount of protein for each assay from specific activity in IMAC purification.

 2)   Table 1 ~ SDS-PAGE in each step will be helpful to many readers.

 3)   Lane 118 and Figure 3

The Mg2+ concentration in the assay mixture set up 5 mM. Concentration dependent manner for ATP over 5 mM will be set up over 5 mM of Mg2+ in Figure 3?

 4)   Figure 2 

Amounts of enzyme was varied between 4 experiments, especially enzyme unit and incubation time etc. It is very difficult to check how different between each assay. Authors will describe the details in the footnote.

 5)   Figure2, 3 and Table 2

Data in Table 2 was calculated from Figure 2 and 3. Authors can combine the data in table 2 in Figure 2 and 3.

 6)   Lane 235 and Figure 8

“the phosphorylated compound production reached 100% (Figure 8C) after… (approx.  12 h).”

In Figure 8, authors checked time course within 8 min. Please indicate pyruvate amounts after 12 h.

7)   Lane 335 “KGP was synthesized at a 1.5 mmol scale”.

 On the other hand,

Lane 337 “KG (1.9 mmoles, 400 mg)”

Lane 347 “as a white powder in 85% yield”

   Calculation of KGP was 1.615 mmoles from these two sentences.

   How calculate did author for chemical yield? The regeneratable PEP was not base for the chemical yield. The description of detail amounts will be preferred. 

 8)   How did author identify that the beta-configuration preferred alpha-configuration.?

The spectrum sounds like contamination of KG, which was excess amount in reaction mixture, in the main product KGP.   

Author Response

Reviewer 2:

 1- Lane 102 ~ “after protein purification and imidazole was removed just before each experiment in order to avoid possible chemical interferences.” Please describe typical specific activity (U/mg) after desalt procedure. Many readers recalculated amount of protein for each assay from specific activity in IMAC purification.

Specific activity was not affected by the desalting procedure. No loss of activity was detected after desalting so specific activity of the final imidazole-free fraction remained the same than after IMAC purification. This point has been clarified in the main text of the manuscript (lanes 104 to 105), and a more detailed desalting procedure explanation has been included in the “Materials and Methods” section (point 3.2.2., “Expression and purification”, lanes xx to xx).

 2- Table 1 ~ SDS-PAGE in each step will be helpful to many readers.

We agree. We have a SDS-PAGE gel to provide we added in the supplementary info part. We are just sorry that the contrast is quite low   

see attached pdf for photo of gel.

Lane 3 contains the enriched KguK protein

3-  Lane 118 and Figure 3 : The Mg2+ concentration in the assay mixture set up 5 mM. Concentration dependent manner for ATP over 5 mM will be set up over 5 mM of Mg2+ in Figure 3?

The data given in Figure 3B were related to activity assays where the Mg2+ concentration was increased to 25 mM. Since the real donor phosphate substrate is the complex Mg-ATP, it is crucial to use higher concentrations of divalent cation than the ATP ones, in order to ensure the right enzyme activity in the assayed conditions.

An explanation of this point has been added in the main test of the revised manuscript (Lanes 117 to 120), and in the footnote of the Figure 3B. In addition, a more detailed description of the experiments shown in Figure 3B has been included in the “Materials and Methods” section (point 3.2.2., “Enzyme activity assays and kinetic studies”, lanes 305 to 308)

4- Figure 2 : Amounts of enzyme was varied between 4 experiments, especially enzyme unit and incubation time etc. It is very difficult to check how different between each assay. Authors will describe the details in the footnote.

Final concentrations of the pure enzyme used in each experiment have been included in Figure 2 footnote, as well as some additional experimental data (Mg2+ and ATP concentrations).

5- Figure2, 3 and Table 2 : Data in Table 2 was calculated from Figure 2 and 3. Authors can combine the data in table 2 in Figure 2 and 3.

We thank you for the suggestion. Nevertheless, we think it is useful to keep table 2 for a better understanding.

6- Lane 235 and Figure 8 : “the phosphorylated compound production reached 100% (Figure 8C) after… (approx.  12 h).” In Figure 8, authors checked time course within 8 min. Please indicate pyruvate amounts after 12 h.

There is a mistake in the time units of Figure 8: it should be hours instead of minutes. This mistake has been corrected, and the additional point at 12h has been added. We therefore have exchanged Fig 8 to a new version

7- Lane 335 “KGP was synthesized at a 1.5 mmol scale”. On the other hand, Lane 337 “KG (1.9 mmoles, 400 mg)” Lane 347 “as a white powder in 85% yield”. Calculation of KGP was 1.615 mmoles from these two sentences. How calculate did author for chemical yield? The regeneratable PEP was not base for the chemical yield. The description of detail amounts will be preferred. 

The 1.5 mmol scale is based on the PEP, which is the limiting substrate (the optimized KG/PEP ratio for maximizing the reaction efficiency was found to be 1.0/0.8). Thus, 1.5 mmol of PEP are totally consumed at the end of the phosphorylation reaction, which we described in the manuscript as “final yield of KGP accumulation in reaction”. Therefore, 1.5 mmols of KGP were formed in the reaction media. After the product purification by Ba2+ precipitation, 1.275 mmoles of KGP barium salt (molecular weight 476.5 g/mol) was recovered as a white powder (0.608g, 85 % yield).

We agree with the reviewer that the way this point is explained in the manuscript could be confusing in some aspects. Therefore, we have changed the definition of “final yield of KGP accumulation in reaction” for “PEP consumption in the reaction” (lanes 240 and 242,  and 353 to 355), and we have added detailed data about the moles and grams of pure compound that were recovered (lanes 362 and 363.

8- How did author identify that the beta-configuration preferred alpha-configuration.?

The configurations were assigned according to a paper published in 1980: “The Structure of Biologically Important Carbohydrates. A 13C Nuclear Magnetic Resonance Study of Tautomeric Equilibria in Several Hexulosonic Acids and Related Compounds”, J. Amer. Chem. Soc., 1980, 102, 2220.

9- The spectrum sounds like contamination of KG, which was excess amount in reaction mixture, in the main product KGP.   

We have noted no KG contamination with KGP, according to the comparison of the NMR spectra. Indeed, the protocol of purification with Ba ions enable to remove the ketoacid, that is why pyruvate was also not observed even though 1.5 mmol are expected after the synthesis.

Round  2

Reviewer 2 Report

Authors made revised manuscript based on he reviewers' comment, so reviewer recommend to publish their manuscript to Molecules.